# Low Conductivity Decay of Sn–0.7Cu–0.2Zn Photovoltaic Ribbons for Solar Cell Application

**DOI:** 10.3390/mi10080550

**Published:** 2019-08-19

**Authors:** Kuan-Jen Chen, Fei-Yi Hung, Truan-Sheng Lui, Lin Hsu

**Affiliations:** 1Instrument Center, National Cheng Kung University, Tainan 701, Taiwan; 2Department of Materials Science and Engineering, National Cheng Kung University, Tainan 701, Taiwan

**Keywords:** Sn–0.7Cu–0.2Zn, photovoltaic ribbon, intermetallic compounds, Zn accumulation layer

## Abstract

The present study applied Sn–0.7Cu–0.2Zn alloy solders to a photovoltaic ribbon. Intermetallic compounds of Cu_6_Sn_5_ and Ag_3_Sn formed at the Cu/solder/Ag interfaces of the module after reflow. Electron probe microanalyzer images showed that a Cu–Zn solid-solution layer (Zn accumulation layer) existed at the Cu/solder interface. After a 72 h current stress, no detectable amounts of Cu_6_Sn_5_ were found. However, a small increase in Ag_3_Sn was found. Compared with a Sn–0.7Cu photovoltaic module, the increase of the intermetallic compounds thickness in the Sn–0.7Cu–0.2Zn photovoltaic module was much smaller. A retard in the growth of the intermetallic compounds caused the series resistance of the module to slightly increase by 9%. A Zn accumulation layer formed at the module interfaces by adding trace Zn to the Sn–0.7Cu solder, retarding the growth of the intermetallic compounds and thus enhancing the lifetime of the photovoltaic module.

## 1. Introduction

In order to achieve the goal of global carbon reduction, countries are actively developing various types of renewable energy. At present, solar energy is one of the most attractive types of energy. An important issue in research on solar photovoltaic (PV) systems is the reduction of the cost of PV ribbons, in addition to the improvement of the photoelectric conversion efficiency of solar cell materials [1,2]. 

In developing low-cost PV ribbons, Sn–Cu alloys are considered promising lead-free alloy solders due to their good weldability and conductivity [3,4]. However, the excessive formation of an intermetallic compound (IMC) at the interface between the Sn–Cu alloy and the Cu substrate during a prolonged heating is problematic [5], and the Sn–Cu alloy has a higher melting point than the Sn–Pb alloy. Our previous report showed that the series resistance of the PV module increased by 52% due to the rapid growth of IMCs and the overconsumption of the Ag electrode after bias for a long time [3]. Note that adding a small amount of Ag or Zn into an Sn–Cu alloy can reduce the above problems [6,7]. Adding Ag into a Sn–Cu solder can reduce the consumption of the Ag electrode in the module after reflow. However, the addition of Ag in the solder increases the module cost and also the proportion of a high-resistance phase in the solder alloy [8]. Considering the cost of the solder, a Zn-containing Sn–Cu solder is examined as a candidate solder to improve the properties of the Sn–Cu alloy. The phase diagram of the Cu–Zn alloy showed that the solid solubility of Zn in Cu can reach 37 wt% at 300 °C [9]. Related studies indicated that Zn atoms will accumulate in the Cu–Zn solid solution alloy at the interface between the solder and Cu substrate and form a Cu–Zn solid solution alloy [6,10]. This Zn accumulation layer could effectively retard the diffusion of Cu atoms into the solder alloy, reducing the growth of IMCs at the interface. 

In the present study, an Sn–0.7Cu alloy solder [3] with 0.2 wt% Zn (Sn–0.7Cu–0.2Zn) is used for the fabrication of a PV ribbon, and the contribution of Zn to the series resistance and interfacial microstructure of the module is investigated.

## 2. Experimental Procedure

A pure Cu ribbon (50 mm × 1.6 mm × 0.2 mm) was immersed in a molten Sn–0.7Cu–0.2Zn alloy solder (400 °C) for 1 s to form a PV ribbon. Then, a fixed contact length (20 mm) of the ribbon was reflowed onto an Ag electrode on a Si solar cell at 300 °C for 5–10 s. A peeling force test was performed to estimate the bonding reliability of the Sn–0.7Cu–0.2Zn/Ag interface. A direct current (DC) of 16 A was applied to the PV module for 72 h to simulate a photo-generated current in the solar cell. The current stress on the solder connections (Cu/solder and solder/Ag) in the test module was defined to be 0.25 A/mm^2^. The detailed testing procedures and the module configuration are described in our previous report [10].

The interfacial microstructure of the PV module was examined using an optical microscope (OM) and scanning electron microscopy (SEM). The diffusive behavior of Cu, Zn, Sn, and Ag elements in the PV module was analyzed by an electron probe microanalyzer (EPMA). The PV module structure (Cu/solder/Ag) was biased from 0 to 16 A in increments of 1 A, and the series resistance of the module was estimated according to Ohm’s law (*V = IR*).

## 3. Results and Discussion

Figure 1 shows interfacial images (Cu/solder/Ag) of the Sn–0.7Cu–0.2Zn PV module after reflow for 5, 7, and 10 s. Cu_6_Sn_5_ and Ag_3_Sn IMCs appeared at the Cu/solder and solder/Ag interfaces, respectively, in all PV modules. The solder matrix was composed of a eutectic structure (β-Sn + Cu_6_Sn_5_ + Cu_5_Zn_8_). Note that the IMC thickness (Cu_6_Sn_5_, Ag_3_Sn) increased with the increasing reflow time, which is related to a static heat-induced (reflowed, hot-dipped) growth reaction [2]. The peel strength of the PV ribbons reflowed onto an Ag electrode on a Si solar cell for various reflow times (5–10 s) is shown in Figure 2. For the reflow time of 5 s, the peeling force of the PV module was lower than a standard value of 100 gf. This was attributed to the smaller IMC growth at the solder/Ag interface, which reduced the bond strength of the PV module. From the statistical results of ImageJ calculations (Figure 2), the thicknesses of the Cu_6_Sn_5_ and Ag_3_Sn IMCs increased with increasing reflow time. This was attributed to the formation of a too small amount of Ag_3_Sn (≤ 1 μm) in the reflow time of 5 s which was insufficient to connect the PV ribbon to the solar cell. Under different reflow conditions (300 °C for 10 s), the IMCs thickness of the Sn–0.7Cu–0.2Zn PV module was lower than that of the Sn–0.7Cu one [3]. This indicated that adding less Zn to the Sn–0.7Cu solder can retard the overgrowth of IMCs in the module.

Considering the bonding strength and residual Ag electrode, the PV module with reflow time of 7 s was subjected to the biasing test.

Figure 3 shows EPMA images of the Cu/Sn–0.7Cu–0.2Zn solder/Ag structure obtained under a given set of reflow conditions. The Cu_6_Sn_5_ compound formed at the Cu/solder interface during the hot-dipping process (Figure 3a). The solder contained a β-Sn matrix and a eutectic structure (β-Sn + Cu_6_Sn_5_ + Ag_3_Sn + (Ag, Cu)_5_Zn_8_) composed of Sn, Cu, Ag, and Zn elements. Note that a higher amount of Zn was present at the solder/Cu interface. According to a previous report [6], the accumulating Zn atoms might be a Cu–Zn solid-solution layer near the Cu metal surface. This Zn accumulation layer could retard the diffusion of Cu into the solder matrix, slowing the growth of IMCs. At the solder/Ag interface (Figure 3b), Ag atoms combined with Sn to form Ag_3_Sn IMC after reflow and also diffused into the solder matrix to form the eutectic structure, which was attributed to reflow-induced thermal diffusion. In addition, a Zn accumulation layer also existed at the solder/Ag interface and could retard the growth of Ag_3_Sn IMC. After biasing for 72 h (Figure 4), a small amount of Ag_3_Sn phase appeared at the Cu/solder interface (Figure 4a), while a Cu_6_Sn_5_ phase was detected at the solder/Ag interface (Figure 4b). This is typical of a diffusion behavior in two different directions and allowed to exclude electromigration, which occurs in a single direction [2]. The growth of Ag_3_Sn and Cu_6_Sn_5_ IMCs was related to an elemental thermomigration. This was attributed to a bias-induced Joule heat which increased the diffusion rate between the atoms, causing the IMCs (Ag_3_Sn and Cu_6_Sn_5_) to migrate to another interface. Compared with the unbiased module, the 72 h-biased module showed increased consumption of the Ag electrode, while the Cu_6_Sn_5_ IMC did not display obvious growth. This result confirmed that a Zn accumulation layer existed at the Cu/solder interface, decreasing the diffusion of Cu atoms and retarding the overgrowth of Cu_6_Sn_5_ IMC.

The measurements of IMCs thickness and series resistance of the PV modules before and after biasing are shown in Figure 5 [3]. The series resistance of an unbiased Sn–0.7Cu–0.2Zn PV module (2.1 × 10^−2^ Ω) was lower than that of an unbiased Sn–0.7Cu module (2.3 × 10^−2^ Ω). This result was attributed to the Sn–0.7Cu–0.2Zn PV module having a thinner Cu_6_Sn_5_ and Ag_3_Sn IMC layer. Adding Zn into the Sn–0.7Cu solder could retard the growth of Cu_6_Sn_5_ IMC, and the shorter reflow time (7 s) decreased the growth of Ag_3_Sn. After biasing for 72 h, the Cu_6_Sn_5_ and Ag_3_Sn IMCs of the Sn–0.7Cu–0.2Zn PV module did not significantly increase compared with those of the Sn–0.7Cu PV module. The growth of Cu_6_Sn_5_ and Ag_3_Sn IMCs in the Sn–0.7Cu–0.2Zn PV module increased by 1% and 34%, respectively, while that in the Sn–0.7Cu PV module increased by 108% and 105%, respectively [3]. In addition, the peeling force of the 72 h-biased Sn–0.7Cu PV module (~30 gf ± 20 gf) was much smaller than that of the reflowed Sn–0.7Cu PV module (~90 gf ± 10 gf), while that of the 72 h-biased Sn–0.7Cu–0.2Zn PV module only dropped to ~160 gf. This indicates that the overgrown IMCs decreased the module resistance and weakened the peeling force of the module. The chemical reaction between the Cu ribbon and the solders mainly dominated the growth behavior of Cu_6_Sn_5_ IMC (Figure 6). Some Zn atoms in the Sn–0.7Cu–0.2Zn solder accumulated at the Cu/solder interface to form the Cu–Zn solid-solution layer during the hot-dip process. These Zn atoms could retard the thermal migration and electromigration of Cu atoms and their reaction with the Sn atoms of the solder. This indicates that the Zn accumulation layer could effectively retard the growth of IMCs, and thus the series resistance of the Sn–0.7Cu–0.2Zn PV module only increased by 9%.

## 4. Conclusions

In present study, a trace amount of Zn was added to Sn–0.7Cu to form a ternary Sn–0.7Cu–0.2Zn solder, which was applied to a PV ribbon. For reflowed PV ribbons, the growth of Ag_3_Sn at the solder/Ag interface was the main factor influencing the bond strength of the PV module and also affected the series resistance of the module. The results of the series resistance measurements showed that the series resistance of the Sn–0.7Cu–0.2Zn PV module was lower than that of the Sn–0.7Cu one before and after biasing. The bias-induced thermomigration and electromigration caused the growth of Cu_6_Sn_5_ and Ag_3_Sn IMCs. The Sn–0.7Cu–0.2Zn PV ribbon better retarded the rise in series resistance of the module (only increased by 9%) due to the formation of a Zn accumulation layer at the Cu/solder/Ag interfaces; this layer effectively retarded IMCs overgrowth in the module, decreasing the series resistance and thus enhancing the service life of the PV module.

## Figures and Tables

**Figure 1 micromachines-10-00550-f001:**
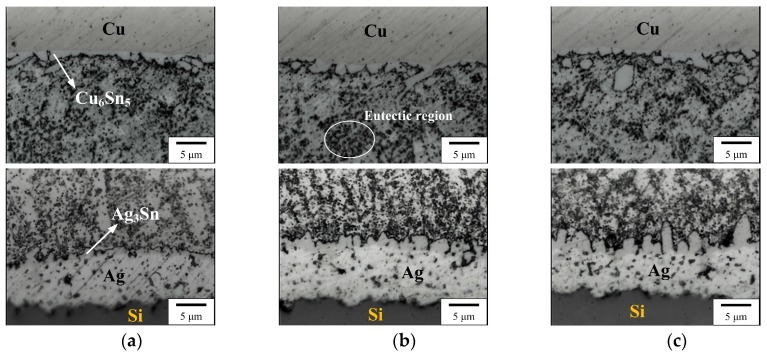
Interfacial images of the structure of a Sn–0.7Cu–0.2Zn photovoltaic (PV) module (Cu/Solder/Ag) obtained after various reflow times: (**a**) 5 s, (**b**) 7 s, and (**c**) 10 s.

**Figure 2 micromachines-10-00550-f002:**
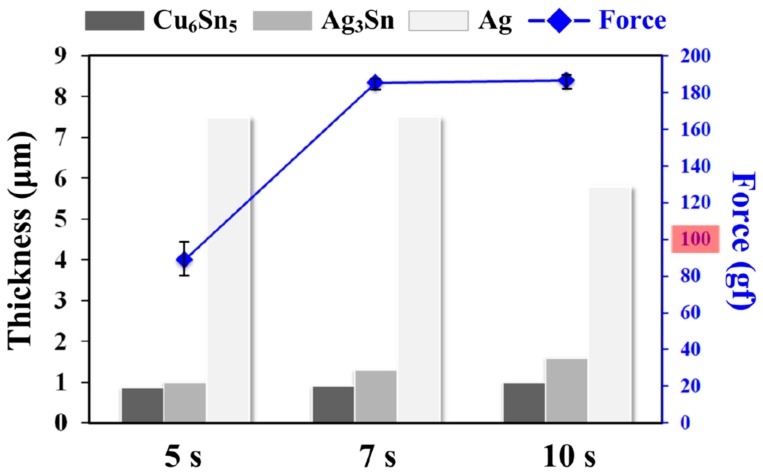
Intermetallic compounds (IMCs) thickness and peel strength of the Sn–0.7Cu–0.2Zn PV module as a function of reflow times.

**Figure 3 micromachines-10-00550-f003:**
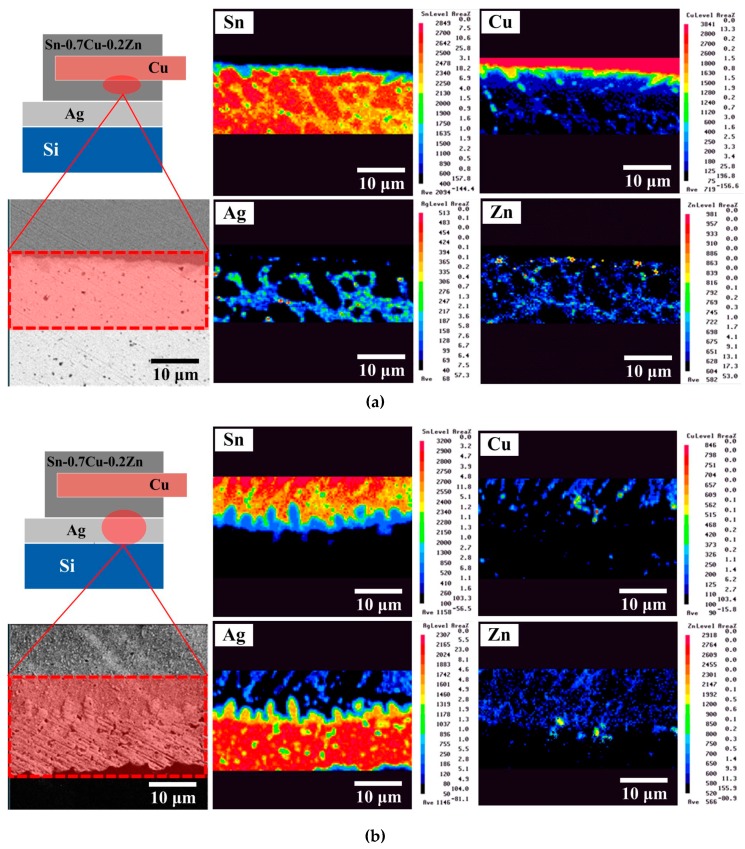
Electron probe microanalyzer (EPMA) images of the unbiased Sn–0.7Cu–0.2Zn PV module at the (**a**) Cu/solder and (**b**) solder/Ag interfacial regions.

**Figure 4 micromachines-10-00550-f004:**
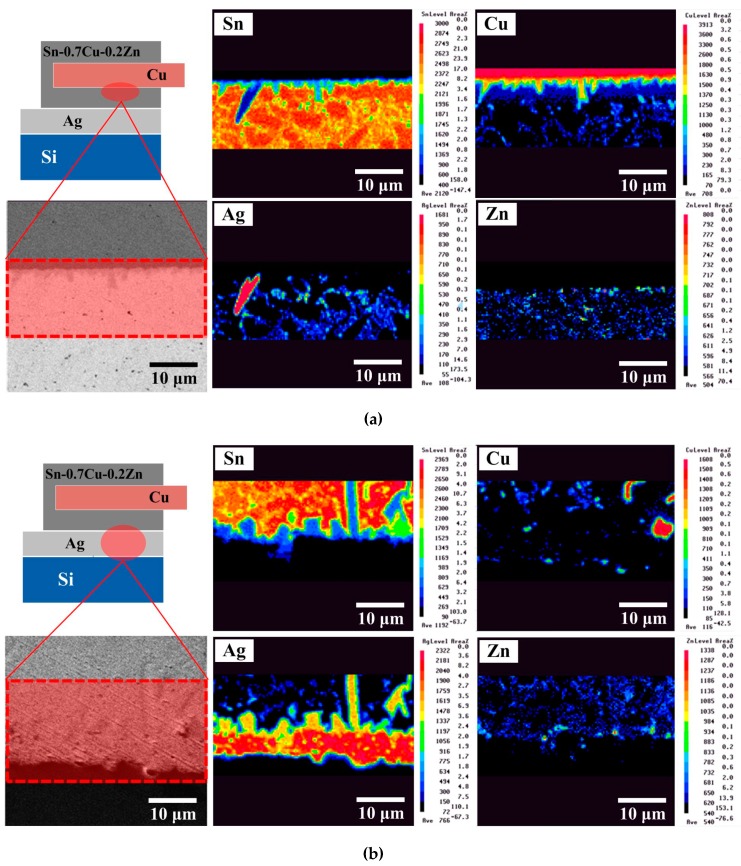
EPMA images of the biased Sn–0.7Cu–0.2Zn PV module at the (**a**) Cu/solder and (**b**) solder/Ag interfacial regions.

**Figure 5 micromachines-10-00550-f005:**
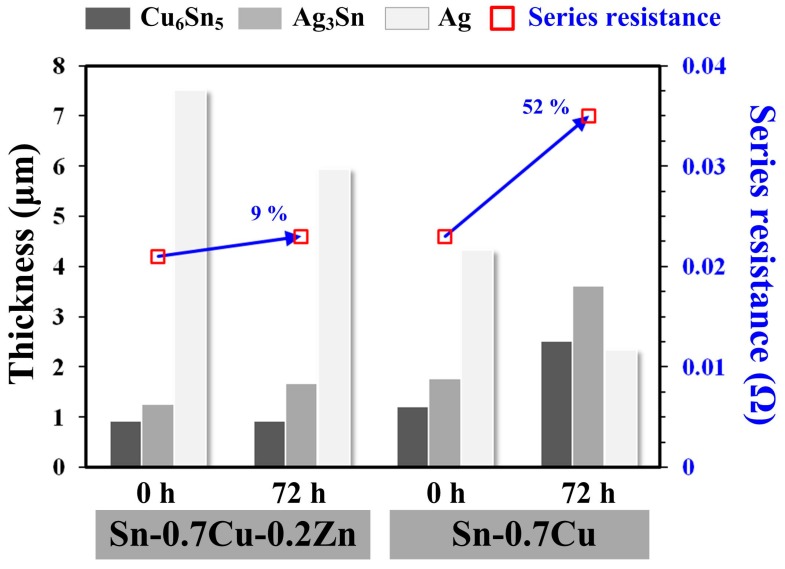
Series resistances of Sn–0.7Cu–0.2Zn and Sn–0.7Cu PV modules as a function of IMCs thickness and bias duration.

**Figure 6 micromachines-10-00550-f006:**
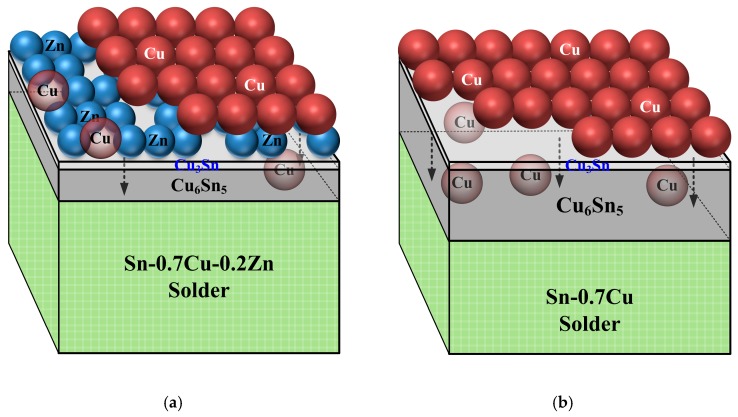
Growth mechanism of Cu_6_Sn_5_ IMC at the Cu/solder interface in the (**a**) Sn–0.7Cu–0.2Zn and (**b**) Sn-0.7Cu PV modules.

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
