# Peer review of "Low Conductivity Decay of Sn–0.7Cu–0.2Zn Photovoltaic Ribbons for Solar Cell Application"

_micromachines, 2019, doi:10.3390/mi10080550_

Round 1
Reviewer 1 Report
The manuscript describes the affect of trace Zn additions to SnAg solders used to "tin" PV module tabing wire. The problem being solved is not well described. Naturally inter-diffusion of elements is both expected and necessary for strong low resistance solder connections. When sufficient concentration near the interface are reached inter metallic structures form.
Through citation or measurement the authors need to demonstrate that these inter-metallic materials actually cause cause resistance increase and weaken the mechanical strength of contacts (a minor decreases in contact strength over time might not be an issue since this would occur after a module is built and deployed when the solder connection is not subject to mechanical stress.
The authors describe a 9% increase in a modules series resistance due to a 16 A forward bias. The reader need to know the module configuration. Specifically the reader needs to know the current stress per unit contact - otherwise the expected gain (in performance) can not be calculated and applied to an arbitrary module configuration. Without this knowledge the experiment lacks reasonable control.
English improvement is needed throughout - for example: abstract third sentence - acronyms need be defined on first use and should not be used in an abstract. The abstract forth sentence is also awkward. It would be better to write "After a 72 hour current stress no detectable quantities of Cu6Sn5 were found. However, a small increase in Ag3Sn was found. Similar problems are found throughout.
Author Response
Response to Reviewer’s Comments
Reviewer 1
The manuscript describes the affect of trace Zn additions to SnAg solders used to "tin" PV module tabing wire. The problem being solved is not well described. Naturally inter-diffusion of elements is both expected and necessary for strong low resistance solder connections. When sufficient concentration near the interface is reached inter metallic structures form.
Our response:
Adding Ag into the Sn-Cu solder can reduce the consumption of Ag electrode in the module after reflow. However, the addition of Ag in the solder increases the module cost, and also increases the proportion of high-resistance phase in the alloy solder [8]. These sentences have been added in the revision paper.
[8] K. J. Chen, F. Y. Hung, T. S. Lui, L. H. Chen, D. W. Qiu, T. L. Chou, “Microstructure and electrical mechanism of Sn–xAg–Cu PV-ribbon for solar cells”, Microelectron. Eng. 116 (2014) 33-39.
Through citation or measurement, the authors need to demonstrate that these inter-metallic materials actually cause resistance increase and weaken the mechanical strength of contacts (a minor decreases in contact strength over time might not be an issue since this would occur after a module is built and deployed when the solder connection is not subject to mechanical stress.
Our response:
According to our previous measurement (Fig a), the peeling force of the 72h-biased Sn-0.7Cu PV module (~30 gf±20 gf) was much lower than the reflowed Sn-0.7Cu PV module (~90 gf±10 gf); while that of 72h-biased Sn-0.7Cu-0.2Zn PV module only dropped to ~160 gf. This indicates that over-growth IMCs decrease the module resistance, and also weaken the peeling force of the module. These sentences have been added in the revision paper.
Fig a Peel force of Sn-0.7Cu and Sn-0.7Cu-0.2Zn obtained after bias for 72 h.
The authors describe a 9% increase in a modules series resistance due to a 16 A forward bias. The reader need to know the module configuration. Specifically the reader needs to know the current stress per unit contact–otherwise the expected gain (in performance) can not be calculated and applied to an arbitrary module configuration. Without this knowledge the experiment lacks reasonable control.
Our response:
The detailed testing procedures and the module configuration can refer to our previous reports [10]. A pure Cu ribbon (50 mm´1.6 mm´0.2 mm) was immersed in molten Sn-0.7Cu-0.2Zn alloy solder to form PV ribbon, and then a fixed contact length (20 mm) of the PV ribbon was reflowed onto an Ag electrode on Si solar cell. The related information has been added in the revision paper.
[10] K. J. Chen, F. Y. Hung, T. S. Lui, L. H. Chen, Y. W. Chen, “A study of green Sn-xZn photovoltaic ribbons for solar cell application”, Sol. Energ. Mater. Sol. C. 143 (2015) 561-566.
English improvement is needed throughout - for example: abstract third sentence - acronyms need be defined on first use and should not be used in an abstract. The abstract forth sentence is also awkward. It would be better to write "After a 72 hour current stress no detectable quantities of Cu6Sn5 were found. However, a small increase in Ag3Sn was found. Similar problems are found throughout.
Our response:
Throughout the manuscript, the English has been revised.

Reviewer 2 Report
The authors present study of photovoltaic ribbons. The topic is interesting and is of practical significance. Overall, the manuscript is well prepared with solid experiment results and discussion. I would like to suggest acceptance of the paper.
I only one minor comment about figure caption. I found the current figure caption is too simplified and not able to clearly explain the figures. In most figures, readers have to guess. For example, Figure 3a has six panels. What are those figures? Please check all the figure caption and make sure enough information is included.
Author Response
Response to Reviewer’s Comments
Reviewer 2
I only one minor comment about figure caption. I found the current figure caption is too simplified and not able to clearly explain the figures. In most figures, readers have to guess. For example, Figure 3a has six panels. What are those figures? Please check all the figure caption and make sure enough information is included.
Our response:
All figures and figure caption have been modified.
Round 2
Reviewer 1 Report
The revisions largely address my initial concerns, except for item #3. The actual current stress applied to the contact remains an unknown. This quantity must be defined! The 16 A current applied to the test module flows through a number of parallel solder connections each having a ribbon bonding area - therefore the current stress on a ribbon contact area basis is: = 16A/(number of parallel cells X number of solder parallel connections per cell X approximate area of one contact). The application of the knowledge gained by the presented work requires the current stress be defined within the paper as it is a central point.
Author Response
Reviewer 1
The revisions largely address my initial concerns, except for item #3. The actual current stress applied to the contact remains an unknown. This quantity must be defined! The 16 A current applied to the test module flows through a number of parallel solder connections each having a ribbon bonding area - therefore the current stress on a ribbon contact area basis is: = 16A/(number of parallel cells ´ number of solder parallel connections per cell ´ approximate area of one contact). The application of the knowledge gained by the presented work requires the current stress be defined within the paper as it is a central point.
Our response:
In present study, a pure Cu ribbon (50 mm´1.6 mm´0.2 mm) was immersed in a molten solder to form PV ribbon. Then, a fixed contact length (20 mm) of the PV ribbon was reflowed onto an Ag electrode on Si solar cell to form PV module. The 16 A current applied to the given module flows through the Cu/solder and solder/Ag interfaces. Therefore, the current stress on the ribbon contact area in the test module is about 0.25 A/mm2 (16A/[1×2×(20mm×1.6mm)]). The current stress on the ribbon contact area has been defined in the revision paper.